# All-cause and cause-specific mortality among individuals imprisoned for driving under the influence of alcohol and drugs in Norway (2000–2016): a retrospective cohort study

Ragnhild Elén Gjulem Jamt [1], Anne Bukten [2,3,4]
Marianne Riksheim Stavseth,[2,3] Stig Tore Bogstrand,[1,5] Torill Tverborgvik[2]

¹Department of Forensic Sciences, Oslo University Hospital, Oslo, Norway
²Norwegian Centre For Addiction Research, University of Oslo, Oslo, Norway
³Section for Clinical Addiction Research, Oslo University Hospital, Oslo, Norway
⁴University College of Norwegian Correctional Service, Lillestrøm, Norway
⁵Department of Public Health Science, University of Oslo, Oslo, Norway

**Correspondence to**
Dr Ragnhild Elén Gjulem Jamt;
rmraej@ous-hf.no

## ABSTRACT

**Aims** To describe all-cause and cause-specific mortality and to investigate factors associated with mortality among individuals imprisoned for driving under the influence (DUI) of alcohol and psychoactive drugs in the Norwegian prison population.

**Design** Retrospective cohort study. The Norwegian prison registry was linked to the Norwegian Cause of Death Registry (2000–2016).

**Setting** Norway.

**Participants/cases** The cohort consisted of 96 856 individuals imprisoned in Norway over a 17-year period obtained from the Norwegian prison registry.

**Primary and secondary outcome measures** Adjusted ORs (aOR) with 95% CI were calculated for death due to any, natural and unnatural causes of death. Analyses were stratified according to DUI convictions: no DUI convictions, only DUI convictions (DUI only), DUI and at least one other drug and alcohol conviction (DUI drug), and DUI and at least one conviction other than drug and alcohol conviction (DUI other).

**Results** In total, 29.3% individuals had one or more imprisonments for DUI. The risk of all-cause mortality was elevated for those convicted for DUI, but only in combination with other types of crimes (DUI drug: aOR=1.5, 95% CI 1.4 to 1.6, DUI other: aOR=1.2, 95% CI 1.1 to 1.4). The risk of death from natural causes was significantly elevated for DUI drug (aOR: 1.8, 95% CI 1.6 to 2.0) and for DUI other (aOR=1.3, 95% CI 1.1 to 1.6). The risk of death from unnatural causes was lower for DUI only (aOR=0.8, 95% CI 0.7 to 0.9) and elevated for DUI drug (aOR=1.5, 95% CI 1.3 to 1.6).

**Conclusions** The risk of all-cause mortality was significantly elevated for those convicted of DUI, but only in combination with other types of crimes.

## STRENGTHS AND LIMITATIONS OF THIS STUDY

⇒ We were able to study the mortality of the entire Norwegian prison population through linkage of mandatory national registries during an observation period of 17 years.
⇒ Linkage of data through unique Personal Identification Number reduces the chances of linkage bias and loss to follow-up.
⇒ The lack of information on background sociodemographics and health variables in the dataset implies that we could not adjust for important pre-existing conditions associated with mortality.
⇒ Data on individuals who served conditional sentences outside prison for driving under the influence were not available.

associated with road traffic injuries and deaths.[1–5] DUI poses a threat to the lives and safety of people who engage in DUI,[6] as well as other road users[7]; however, the research regarding future outcomes of people convicted of DUI of alcohol and drugs is both limited and somewhat outdated.[8 9]

Drivers who engage in DUI have been characterised by serious health problems such as substance use and dependence, major depressive disorders, anxiety disorders, hyperactivity disorders and risk-taking dispositions.[8] People arrested for DUI exhibit similar patterns of health problems and mortality as found among people with alcohol use disorders.[10] Studies by Gjerde *et al* and Ruud *et al* found that between 35% and 60% of those arrested for DUI in Norway had an alcohol use disorder.[11–13] In Germany, more than 80% of persons arrested for DUI of alcohol had a blood alcohol concentration (BAC) higher than 1.9 g/dL and could be characterised as having alcohol use disorders.[14] People convicted of DUI have also been characterised

## INTRODUCTION

The literature on driving under the influence (DUI) of psychoactive substances (alcohol, psychoactive medicinal drugs, and illicit drugs) has mainly revolved around topics such as the prevalence of DUI and the risks

by lower levels of education and higher rates of unemployment, disability pensions and divorce compared with the general population.[15]

Norway has practised a per se law for DUI of alcohol since 1936. Since 2001, the legal limit in Norway for DUI of alcohol in blood is 0.02 g/dL. Graded sanctioning is practised; a driver is sentenced to conditioned imprisonment for driving with BACs between 0.05 g/dL and 0.12 g/dL, and unconditioned imprisonment for BACs above 0.12 g/dL.[16] Per se limits for 28 psychoactive drugs were implemented in 2012/2016 corresponding to a BAC of 0.02 g/dL.[17] These psychoactive drugs includes both medicinal and illicit drugs such as benzodiazepines, cannabis, opioids and psychostimulants (amphetamines, cocaine). In addition, limits for graded sanction corresponding to BACs of 0.05 and 1.2 g/dL for 24 of these drugs have been implemented.[18]

The most recent roadside survey in Norway was performed in 2016/2017 and included 5034 drivers.[19] The total prevalence of alcohol and psychoactive drugs among random drivers in this study was around 5%. While the prevalence of drivers with a BAC above the legal limit of 0.02 g/dL was 0.2%, the total prevalence of psychoactive medicinal drugs and of illicit drugs were 3.0% and 1.7%, respectively. The prevalence of drugs in concentrations above the legislative limits was 1.1% for medicinal drugs and 0.7% for illicit drugs. Hence, DUI of alcohol is less common in Norway. Legislation on DUI, including strong law enforcement and strict punishments over many years, is likely to have contributed to the low numbers of DUI in Norway.[20] However, the prevalence of psychoactive substances among drivers arrested on suspicion of DUI and among drivers involved in road traffic crashes is high.[21 22] A paper reviewing the risk of serious road traffic crashes while engaging in DUI found higher risks in Norwegian and Finnish studies than in studies from other countries.[23] It has been suggested that in countries with a strong law enforcement and strict punishments for DUI, people DUI are a more selected group with higher incidence of substance use disorders (SUDs) along with higher levels of sensation-seeking and risk-taking behaviour.[21] It is furthermore suggested that people engaging in risky driving behaviour might be less affected by threats of punishment.[24]

Norway is characterised by having low rates of imprisonment. In 2021, the prison population rate per 100 000 of the national population was 56 in Norway, compared with 629 in the US and 131 in the UK.[25] The Norwegian penal system aim at rehabilitation in that the sentence is recognised as the punishment, and universal healthcare, including drug treatment, is to be provided in prison as it is outside of prison.[26]

Norwegian prison units are public and are located all over Norway. This prison organisation allows most prisoners to preserve geographical closeness to friends and family. In 2020, 18% of all unconditional sentences lasted less than 30 days and 76% lasted up to 1 year.[27] Women constitute a minority in Norwegian prisons, with an annual proportion of approximately 5%.[28]

In 2019, 21% of all prison sentences in Norway included at least one DUI-offence.[29] Several studies have demonstrated that the prison population has a greater risk of premature death compared with the general population, particularly in the short-term period following release from prison.[30–33] A study based on the Norwegian police registration system showed that the risk of premature death is particularly high among individuals arrested for alcohol and drug related crimes, compared with both the general population and individuals arrested for other criminal offences.[9]

Although there are similarities between people convicted of DUI and individuals with alcohol use disorders in terms of health problems, the entire group of people convicted of DUI is large and heterogeneous, and therefore more specific knowledge on the risk factors and mortality is warranted for this group.[10]

Using data on all individuals imprisoned in Norway in the period 2000–2016, we defined three subcategories of the DUI group in order to examine combinations of DUI convictions and other convictions in relation to mortality. The aims of this paper were to:

1. describe the different groups of individuals imprisoned for DUI in the Norwegian prison population,
2. describe all-cause and cause-specific mortality in the DUI groups and compare with individuals imprisoned for other crimes than DUI, and
3. investigate factors associated with natural and unnatural mortality in the prison population.

## METHODS
### Design, study population and data sources
In this retrospective cohort study, we used data from the Norwegian Prison Release Study (nPRIS),[34] including all persons imprisoned in Norway over a 17-year period (1 January 2000, until 31 December 2016). The nPRIS-cohort constitutes 114 745 individuals collected from the Norwegian Prison Registry. The registry serves administrative and statistical purposes and includes personal data on all persons imprisoned in Norway, including age, sex, convictions and sentences.[26] The nPRIS-cohort includes individuals who started serving a prison sentence, had an ongoing prison sentences and individuals who were released from prison during the observation period. Information on any prison sentences prior to the start of the observation period started was not available.

Data from the nPRIS-cohort were linked to the Norwegian Cause of Death Register (NCoDR) through the unique 11-digit Norwegian Personal Identification Number (PIN). The PIN is given to all persons who are born in Norway, have a valid residence permit for at least 6 months or have officially moved to Norway. The NCoDR includes death certificates reported by medical doctors after examination of the deceased. The NCoDR includes information on time of death along with the underlying cause of death (the disease or injury which initiated the chain of morbid events leading directly to death) and

**Table 1** Categorisation of causes of death according to ICD-10 codes

| Cause of death | ICD-10 code |
|---|---|
| Cancer | C00-C97 |
| Circulatory | I00-I99 |
| Respiratory | J00-J99 |
| Digestive | K00-K69, K71-K93 |
| Alcoholic liver disease | K70 |
| Other natural | A, B, D-H, L-Q |
| Natural causes of death | All codes listed above |
| Transport-related | V, Y85 |
| Intoxication* | |
| Alcohol-related | F10, X45, Y15 |
| Drug-related | F11, F12, F14-16, F19, X41, X42, X44, Y11, Y12, Y14 |
| Suicide | X60-X84 |
| Homicide | X85-X99, Y00-Y09 |
| Other accident related | W, X2, Y2 |
| Unnatural causes of death | All codes listed above (excluding natural causes) |
| Unexplained | R95-R99 |
| Unknown cause | No ICD-10 code recorded in NCoDR |

*Alcohol-related and drug-related causes combined.
ICD-10, International Classification of Diseases, 10th revision; NCoDR, Norwegian Cause of Death Registry.

the immediate causes of death (the terminal event or complication present at the time of death).[35] All causes of death are coded using the International Classification of Diseases, 10th revision (ICD-10).[36]

In total, 19 140 individuals were excluded from the study cohort due to lack of a valid PIN. Hence, the study population consisted of 96 856 individuals, including both those who died and those who lived during the observation period, contributing to 167 068 releases.

## Measures
Individuals in the cohort who had one or more convictions for DUI were divided into three subgroups: *DUI-only* as those having no other convictions but DUIs during the observation period; *DUI-drug* as those having both DUI and at least one other drug and alcohol conviction during the observation period; and *DUI-other* as those having DUI and at least one other conviction other than drug and alcohol conviction during the observation period.

We categorised all causes of death as either natural or unnatural, as described in table 1. Natural causes were defined as ICD-10 chapters A through Q, while unnatural causes were defined as ICD-10 chapters V through Y. Drug-related and alcohol-related causes (F10–F19) were defined as unnatural causes of death (table 1).

We classified types of convictions according to the nine-group classification used for Statistics Norway's official crime statistics.[37] In their statistics, DUI offences are included in the category 'Drug and alcohol offences', but as we wanted to study this offence in particular, we constructed a subgroup with DUI convictions exclusively. Thus, the 10-group classification used in this paper were: Property theft; Other offences for profit; Criminal damage; Drug and alcohol offences (not including DUI); Public order and integrity violations; Sexual offences; Traffic offences; Violence and maltreatment; Other offences; and DUI. One imprisonment can include multiple convictions. Convictions were aggregated across the study period, and dummy coded as 'convicted of' (1) or 'not convicted of' (0).

## Statistical analyses
All analyses were performed using SPSS, V.26. We made group comparisons using Student's t-test or Mann-Whitney U-test for continuous data and $\chi^2$-test for categorical data. Kruskal-Wallis test was used for testing distribution between subgroups.

The associations between mortality and DUI convictions were examined using logistic regression. The coefficients were interpreted as ORs with 95% CI. Individuals who died due to natural causes differed significantly from those who died due to unnatural causes. Therefore, separate logistic regression models were fitted. The three regression models were defined with the following outcome of interest: (1) all-cause mortality (n=8053), (2) death due to natural causes (n=3379) and (3) death due to unnatural causes (n=4139). To avoid competing risks, unknown and unnatural cause deaths were excluded from analysis in the second regression model, and unknown and natural cause deaths were excluded from analysis in the third model. Univariate regression models were fitted, and statistically significant covariates (p<0.05) were included in the multivariable analysis.

## Patient and public involvement
Patients and/or the public were not involved in the design, or conduct, or reporting, or dissemination plans of this research.

## RESULTS
### Participants
The cohort consisted of 96 856 individuals of which 28 393 individuals (29.3%) were convicted of any DUI (table 2). Among these, 8898 (9.2%) were only convicted of DUI (DUI-only), 16 159 (16.7%) were convicted of DUI and at least one drug and alcohol offence (DUI-drug), and 3336 (3.4%) were convicted of DUI and at least one crime other than drug or alcohol related offences (DUI-other) during the observation period (table 2). The DUI-only group had the highest proportion of women (13.7%) and were the oldest at last incarceration (mean age 38.4 years). There were significant differences in the mean

 

**Table 2** Background characteristics of the study population (n=96 856), 1 January 2000–31 December 2016

| | No DUI | | DUI only | | DUI drug | | DUI other | |
| --- | --- | --- | --- | --- | --- | --- | --- | --- |
| | n | % | n | % | n | % | n | % |
| Total | 68 463 | 70.7 | 8898 | 9.2 | 16 159 | 16.7 | 3336 | 3.4 |
| Sex* | | | | | | | | |
| Men | 61 171 | 89.3 | 7680 | 86.3 | 14 870 | 92.0 | 3155 | 94.6 |
| Women | 7275 | 10.6 | 1218 | 13.7 | 1289 | 8.0 | 181 | 5.4 |
| Age at last conviction†‡ | 34.6 (11.8) | | 38.4 (14.1) | | 36.2 (11.4) | | 36.6 (13.4) | |
| Number of conviction‡§ | 1.6 (1.4) | | 1.1 (0.3) | | 2.7 (2.6) | | 1.6 (0.9) | |
| Convictions¶ | | | | | | | | |
| Property theft | 9974 | 14.6 | – | – | 5063 | 31.3 | 351 | 10.5 |
| Other offences for profit | 16 335 | 23.9 | – | – | 4197 | 26.0 | 358 | 10.7 |
| Criminal damage | 4045 | 5.9 | – | – | 2102 | 13.0 | 171 | 5.1 |
| Drug and alcohol offences** | 28 505 | 41.6 | – | – | 16 159 | 100 | – | – |
| Public order/integrity | 15 379 | 22.5 | – | – | 5673 | 35.1 | 652 | 19.5 |
| Sexual offences | 5362 | 7.8 | – | – | 333 | 2.1 | 124 | 3.7 |
| Traffic offences | 3648 | 5.3 | – | – | 5372 | 33.2 | 1932 | 57.9 |
| Violence and maltreatment | 25 597 | 37.4 | – | – | 4932 | 30.8 | 1061 | 31.8 |
| Other offences | 2278 | 3.3 | – | – | 947 | 5.9 | 89 | 2.7 |
| Driving under the influence | – | – | 8898 | 100 | 16 159 | 100 | 3336 | 100 |
| *1DUI conviction* | – | – | *8657* | *97.3* | *12 755* | *78.9* | *3032* | *90.9* |
| *2+DUI convictions* | – | – | *241* | *2.7* | *3404* | *21.1* | *304* | *9.1* |
| Deceased | 5064 | 7.4 | 825 | 9.3 | 1825 | 11.3 | 339 | 10.2 |

*17 persons missing information on sex.
†42 persons missing information on age.
‡Continuous measures presented with mean (SD).
§During the study period.
¶During the study period. Note:a person may have several convictions per incarceration and several incarcerations.
**Not including DUI.

age at last incarceration between the groups (results not shown).

Approximately one-third of the total cohort (n=30 506) had more than one imprisonment during the observation period. In the DUI-drug group, almost 60% had more than one imprisonment; 21% had more than one imprisonment for DUI. Only 5.5% of the DUI-only group had multiple incarcerations for DUI during the study period. The overall proportion of deceased among those convicted of any DUI was 10.5% compared with 7.4% among the non-DUI offenders. The proportion of deceased was highest in the DUI-drug group (11.3%) (table 2).

### Causes of death

Deaths among those convicted of DUI-only were mostly due to natural causes (58.9%; table 3). Diseases of the circulatory system and cancer were the predominant natural causes of death in all groups. A higher proportion of death due to alcoholic liver disease was found among those convicted of DUI only (5.2%). Those convicted of DUI only had the lowest proportion of unnatural causes of death (33.0%).

Deaths among those convicted of DUI drug were more often due to unnatural causes (54.1%), as was the case among those convicted of other crimes than DUI (53.9%). Intoxication (drug or alcohol related) accounted for the largest proportion of the unnatural causes of death in all groups. Among those convicted of DUI drug, the vast majority of the deaths caused by intoxications were drug related.

About 5% of mortalities in the DUI-drug and DUI-other groups were transport related. The proportion in the DUI-only group was 1.6%.

The DUI-only group was in general older at time of death than the other groups. Mean age at death was 61 years among DUI only, but only 48 years in the DUI-drug group (online supplemental table 1). Overall, the non-DUI offenders had the lowest mean age at death of 46 years.

**Table 3** Cause of death (n, %) stratified by DUI groups

| | No DUI | | DUI | | | | | |
| | | | DUI only | | DUI drug | | DUI other | |
| Cause of death | n | % | n | % | n | % | n | % |
|---|---|---|---|---|---|---|---|---|
| Cancer | 640 | 12.6 | 144 | 17.5 | 214 | 11.7 | 46 | 13.6 |
| Circulatory | 620 | 12.2 | 158 | 19.1 | 234 | 12.8 | 62 | 18.3 |
| Respiratory | 215 | 4.2 | 55 | 6.7 | 86 | 4.7 | 12 | 3.5 |
| Digestive | 98 | 1.9 | 22 | 2.7 | 43 | 2.4 | 7 | 2.1 |
| Alcoholic liver disease | 105 | 2.1 | 43 | 5.2 | 56 | 3.1 | 16 | 4.7 |
| Other natural | 325 | 6.4 | 64 | 7.8 | 94 | 5.2 | 20 | 5.9 |
| **Total natural cause** | **2003** | **39.6** | **486** | **58.9** | **727** | **39.8** | **163** | **48.1** |
| Transport-related | 224 | 4.4 | 13 | 1.6 | 90 | 4.9 | 17 | 5.0 |
| Intoxication* | 1606 | 31.8 | 164 | 19.9 | 601 | 32.9 | 70 | 20.6 |
| *Alcohol-related* | *170* | *3.4* | *75* | *9.1* | *97* | *5.3* | *31* | *9.1* |
| *Drug-related* | *1436* | *28.4* | *89* | *10.8* | *504* | *27.6* | *39* | *11.5* |
| Suicide | 569 | 11.2 | 54 | 6.5 | 160 | 8.8 | 28 | 8.3 |
| Homicide | 69 | 1.4 | – | – | 31 | 1.7 | – | – |
| Other accident*related | 262 | 5.2 | 39 | 4.7 | 106 | 5.8 | 31 | 9.1 |
| **Total unnatural cause** | **2730** | **53.9** | **272** | **33.0** | **988** | **54.1** | **149** | **44.0** |
| Unexplained | 140 | 2.8 | 35 | 4.2 | 52 | 2.8 | 12 | 3.5 |
| Unknown cause | 191 | 3.8 | 32 | 3.9 | 58 | 3.2 | 15 | 4.4 |
| **Total deceased** | **5064** | **100** | **825** | **100** | **1825** | **100** | **339** | **100** |

- Cells with n < 5 are not displayed.
*Alcohol-related and drug-related combined.

## Factors associated with mortality

After adjusting for relevant covariates, being male and increasing age at last incarceration were factors significantly associated with elevated risk of death from all causes: Sex: (aOR: 1.2, CI 1.1 to 1.3); Age: (aOR: 1.1, CI 1.1 to 1.1). This was also the case for natural causes: Sex: (aOR: 1.2, CI 1.0 to 1.3); Age: (aOR: 1.1, CI 1.1 to 1.1), and unnatural causes: Sex: (aOR: 1.1, CI 1.0 to 1.2); Age: (aOR: 1.0, CI 1.0 to 1.0) (table 4). Number of convictions was not significantly associated with death from all causes

**Table 4** Summary of the logistic regression models for all-cause, natural and unnatural causes of death (for full table, see online supplemental table 2)

| | All cause of death, n=8053 | | Natural causes of death, n=3379 | | Unatural causes of death, n=4139 | |
| | aOR* (95% CI) | P value | aOR* (95% CI) | P value | aOR* (95% CI) | P value |
|---|---|---|---|---|---|---|
| Sex (female ref.) | 1.22 (1.12 to 1.32) | **<0.001** | 1.18 (1.04 to 1.33) | **0.011** | 1.11 (1.00 to 1.24) | **0.063** |
| Age† | 1.06 (1.05 to 1.06) | **<0.001** | 1.11 (1.10 to 1.11) | **<0.001** | 1.01 (1.01 to 1.02) | **<0.001** |
| No. of convictions | 0.99 (0.97 to 1.00) | **0.092** | 0.85 (0.83 to 0.88) | **<0.001** | 1.06 (1.04 to 1.08) | **<0.001** |
| Conviction (no dui ref.): | – | | – | | – | |
| DUI-only | 0.99 (0.91 to 1.07) | **0.916** | 1.11 (0.99 to 1.24) | **0.066** | 0.78 (0.69 to 0.89) | **<0.001** |
| DUI-drug | 1.51 (1.44 to 1.63) | **<0.001** | 1.80 (1.64 to 1.98) | **<0.001** | 1.45 (1.34 to 1.57) | **<0.001** |
| DUI-other | 1.23 (1.09 to 1.38) | **<0.001** | 1.34 (1.12 to 1.60) | **0.001** | 1.13 (0.95 to 1.34) | **0.164** |

Estimates are given as adjusted OR (aOR) with corresponding 95% CIs. The level of statistical significans was p≤0.05.
*Adjusted for covariates significant in crude analyses.
†Age at last incarceratinon.
aOR, adjusted OR; DUI, driving under the influence.

(aOR: 0.99, CI 0.97 to 1.00). However, the variable was a significant protective factor on death from natural causes (aOR: 0.9, CI 0.8 to 0.9) and was a significant risk factor on death from unnatural causes (aOR: 1.1, CI 1.0 to 1.1).

Being convicted of DUI and other crimes significantly elevates the risk of death from all causes (DUI-drug: aOR 1.5, CI 1.4 to 1.6, DUI-other: aOR 1.2, CI 1.1 to 1.4). The same was true for death from natural causes (DUI-drug: aOR 1.8, CI 1.6 to 2.0; DUI-other: aOR 1.3, CI 1.1 to 1.6). While being convicted of DUI and one or more drug and alcohol related crimes increased the risk of dying from unnatural causes (aOR 1.5, CI 1.3 to 1.6), being convicted of DUI only had a significant protective effect on the risk of dying from unnatural causes (aOR 0.8, CI 0.7 to 0.9).

## DISCUSSION

In this retrospective cohort study of mortality among people imprisoned for DUI in Norway, we found the risk of all-cause mortality to be significantly elevated for those convicted of DUI, but only in combination with other types of crimes, and especially in combination with other drug and alcohol offences. Being convicted of no other crime than DUI had a protective effect on unnatural cause of death after adjusting for age, sex and number of previous convictions.

Psychiatric disorders and SUDs are highly prevalent in the prison population,[38 39] and a large proportion of people imprisoned for DUI are likely to have some degree of SUD and/or mental health problems.[8] In our study, almost half of the total cohort had one or more drug and alcohol convictions and 30% had at least one DUI conviction. Among those convicted of DUI, 60% had additional convictions for other drug and alcohol offences.

Adjusted for several other factors, individuals with a history of both DUI and other drug and alcohol convictions had the highest risk of mortality due to both natural and unnatural causes. The most common unnatural causes were intoxication and suicide and the most common natural causes were diseases of the circulatory system and cancer. This is in line with previous research findings on the most dominant causes of death among people convicted of DUI.[10 40] The high mortality from natural causes might indicate unhealthier lifestyles in aspects other than just alcohol and drug use, like smoking, physical inactivity and obesity.[41 42]

Many DUI offenders also have a history of convictions for other types of crime in addition to the DUI offence, and the tendency to relapse into DUI is well known.[43–45] Previous research has identified some characteristics of individuals who relapse into DUI. In American, Finish and Swedish studies, DUI relapse has been associated with factors such as being male, young age, mental health disorders, high concentrations of alcohol and drugs at the time of the arrest, and previous DUI convictions.[44 46–48] Particularly, those DUI of drugs had high rates of rearrests.[44 48] The results of the present study revealed that those convicted of both DUI and other drug and alcohol offences had on average the highest number of incarcerations during the observation period (2.7 incarcerations) and the largest proportion of individuals with

two or more convictions for DUI (21.1%) compared with the two other DUI groups. The proportion of younger males was higher in the DUI-drug group compared with the DUI-only group and about the same as in the DUI-other group.

Impinen et al suggest that an over-representation of unnatural causes of death might be an indicator of a risk-taking lifestyle.[10] Sensation-seeking behaviour is a personality trait that has been associated with engaging in unusual and risky acts, fast and reckless driving, and alcohol and drug use.[49 50] Sensation-seeking behaviour has also been shown to decline with age and is usually more pronounced among males.[51]

The mean age at death was low for the whole cohort and particularly low for death due to unnatural causes. As expected, higher age was significantly associated with increased risk of death. However, when stratifying on cause of death (natural and unnatural causes), the effect of increasing age at last incarceration became more pronounced for the risk of death from natural causes. In a study by Nordentoft et al,[52] high excess mortality due to somatic illnesses was found among patients with psychiatric disorders and SUDs. The excess mortality was highest among patients with SUDs who died 20 years earlier than the general population.

It is evident from the present study that people convicted of DUI are a heterogenic group with those only imprisoned for DUI standing out from both the other DUI groups and from those imprisoned for other crimes than DUI. We found that the 'DUI only'-group were older, had fewer incarcerations, higher proportion of women and were at significantly lower risk of death from unnatural causes. Previous research have identified socioeconomic status, civil status, pre-exciting health conditions, employment or education and living conditions to significantly predict the mortality in the prison population.[53 54] The fact that the DUI-only group were at lower risk of death from unnatural causes compared with all other studied groups in the present study could for instance be related to factors like higher socioeconomic status and a lower prevalence of pre-exciting drug dependence in this group. Another point to make notice of is the selection of the DUI-only group in this study which is based solely on DUI convictions. This indicates that the individuals of the DUI-only group commits less crime and thereby have fewer incarcerations, which in turn is protective on health and mortality.[55] The prison context may offer an important opportunity for interventions in terms of SUD treatment. A Norwegian study on DUI among patients in opioid agonist treatment (OAT) found a 40% reduction in DUI convictions while in treatment. However, patients with two or more convictions for DUI before entering treatment had higher odds for further convictions of DUI during OAT compared with patients having no DUI convictions during the pretreatment period.[56] Multiple DUI convictions might thus be viewed as a proxy for the severity of their substance use.

Attempting to combat DUI recidivism and to lower the social and economic costs of incarceration various DUI prevention programmes have been developed internationally. An example is the *Intensive supervision programs*.[57] These programmes include home confinement with activities such as screening and assessment of offenders' substance

abuse problems, treatment and education related to DUI behaviour, electronic monitoring of offender movements and use of ignition interlock devices or other restrictions on driving. At present, the assessment of the effectiveness of DUI prevention programmes is inconclusive. However, there is some evidence to support the effectiveness of programmes using intensive supervision and education.[58] In Norway, the programme consists of education, individual and group therapy sessions and investigations of the offenders' treatment needs, but is only available for individuals serving conditional sentences, and thus disqualifying people with more severe SUDs typically serving unconditional sentences.

The results of the present study should be interpreted with certain limitations in mind. The lack of information on background sociodemographics and health variables in the dataset implies that we could not adjust for important pre-existing conditions and factors associated with mortality. It is likely that such factors also would have affected the findings in the present study if the information had been available. Data on individuals who served conditional sentences and served sentences with electronic monitoring were not available for the present study. Thus, our data contain a selection bias as we only have data on the most severe cases of legal offences who likely have a greater disease burden and higher mortality. A strength of this study is that we were able to study the mortality of the entire Norwegian prison population through linkage of mandatory national registries during an observation period of 17 years. The linkage of datasets through unique PINs reduces the chances of linkage bias and loss to follow-up is rare. Furthermore, the classification of all deaths is in accordance with the ICD-10 criteria and cause of death are reported by a clinician according to individual ICD codes, reducing the chances of information bias. The NCoDR is a reliable source of data on causes of death with combined degree of coverage and completeness of 98% of the deaths in Norway.[35]

Systematic screening with respect drug and alcohol use disorders is a critically important first step to identify individuals who should be offered treatment during incarceration. However, validated screening tools are scarcely implemented in European prisons.[59] The results from our study imply that in places where systematic screening is not routinely carried out data on current and prior offence(s) a person is sentenced to prison for (eg, DUI, other alcohol and drug offences) might serve as an indicator of harmful alcohol and drug use. These data are readily available to the correctional service staff, and can be used as means for further investigations using validated screening tools to properly identify people with potential harmful alcohol and drug use. Moreover, when risk has been identified, provision of evidence-based treatment and harm reduction services like intensive supervision and education programmes must be provided for those in need.

**Acknowledgements** We thank the Directorate of the Norwegian Correctional Services and the Norwegian Cause of Death Registry for valuable assistance regarding registry data.

**Contributors** REGJ: conceptualisation, formal analysis, investigation, methodology, writing – original draft. AB: conceptualisation, investigation, methodology, supervision, writing – review and editing. MRS: conceptualisation, formal analysis, investigation, methodology, supervision, writing – review and editing. STB: conceptualisation, methodology, supervision, writing – review and editing. TT: conceptualisation, formal analysis, methodology, supervision, writing – review and editing. REGJ is guarantor. All authors have approved the final article.

**Funding** This work was supported by the South-Eastern Norway Regional Health Authority: grant number: 2019091 and the Norwegian Research Council: grant number: 301535.

**Competing interests** None declared.

**Patient and public involvement** Patients and/or the public were not involved in the design, or conduct, or reporting, or dissemination plans of this research.

**Patient consent for publication** Not applicable.

**Ethics approval** This study involves human participants and was approved by Regional Committees for Medical and Health Research Ethics, Region South-East Norway, reference number 2012/140 and the Norwegian Directorate for Correctional Services. The study was approved exempt from informed consent.

**Provenance and peer review** Not commissioned; externally peer reviewed.

**Data availability statement** No data are available. The data for the present study are not available for sharing due to ethical considerations.

**ORCID iDs**
Ragnhild Elén Gjulem Jamt http://orcid.org/0000-0002-8619-340X
Anne Bukten http://orcid.org/0000-0003-0285-5339

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
