## [Reviewer comments · BMJ Open]

ARTICLE DETAILS

TITLE (PROVISIONAL)	All-cause and cause-specific mortality among individuals imprisoned for Driving under the Influence of alcohol and drugs in Norway (2000-2016): a retrospective cohort study
AUTHORS	Jamt, Ragnhild; Bukten, Anne; Stavseth, Marianne; Bogstrand, Stig Tore; Tverborgvik, Torill

VERSION 1 – REVIEW

REVIEWER	Fell, James NORC at the University of Chicago, Economics, Justice & Society
REVIEW RETURNED	05-Sep-2023

GENERAL COMMENTS	Nice study. I just have a few questions for your response: (1) In Table 2 on page 10, is the age at last incarceration statistically different for each of the DUI groups? Was that tested? (2) The DUI-only group had a protective effect on the risk of dying from external causes. Can you discuss why that would be? Give reasons? Was it because there were a greater proportion of females in the group? (3) How can your results be applied in practice? Research to practice. Other than screening offenders, what other recommendations do you have?
--

REVIEWER	Schumann, Jennifer
REVIEW RETURNED	13-Sep-2023

GENERAL COMMENTS	A well written paper with novel linkage of datasets, providing useful data on DUI in Norway. Annotations provided in the pdf indicate where clarification is needed, as well as some minor grammatical corrections.
---

VERSION 1 – AUTHOR RESPONSE

Reviewer: 1

Dr. James Fell, NORC at the University of Chicago

Comments to the Author:

Nice study. I just have a few questions for your response:

(1) In Table 2 on page 10, is the age at last incarceration statistically different for each of the DUI groups? Was that tested?

Thank you for this comment. Age at last incarceration was tested and a statistical significant difference was found between the groups. We have added this information to the text in the first paragraph of the Results section.

(2) The DUI-only group had a protective effect on the risk of dying from external causes. Can you discuss why that would be? Give reasons? Was it because there were a greater proportion of females in the group?

This is an interesting topic that we would have liked to explore in more details. There are several factors that are likely to have an effect on our finding, like socioeconomic status, employment/unemployment, civil status, pre-existing medical conditions etc. that we lack the information on in our data and thereby could not adjust for in the logistic regression model. However, we have elaborated on this, and suggest how such factors could have affected our results in the 6th paragraph of the Discussion section..

(3) How can your results be applied in practice? Research to practice. Other than screening offenders, what other recommendations do you have?

We have added a suggestion in the last paragraph of the Discussion section.

Reviewer: 2

PLEASE NOTE: Reviewer 2 has also provided comments in a PDF document that is attached to this email.

Dr. Jennifer Schumann, Monash University

Comments to the Author:

A well written paper with novel linkage of datasets, providing useful data on DUI in Norway. Annotations provided in the pdf indicate where clarification is needed, as well as some minor grammatical corrections.

Reply to comments:

P4, line 5: convicted of

This is corrected in the revised manuscript.

P4, line 7: what is “drug and alcohol crime”

We have change the term to “drug and alcohol offences”

P4, line 18: spell out PIN

We have spelled out personal identification number (PIN).

P5, line 5: ...driving under the influence (DUI) of psychoactive...

This is corrected in the revised manuscript.

P5, line 7: remove 'DUI' from here so it just refers to the psychoactive substances being referred to

This is corrected in the revised manuscript.

P5, line 12: people who are convicted of DUI? Or anyone driving under the influence?

Convicted of DUI. This has been corrected in the manuscript.

P5, line 12: alcohol and other drugs, rather than alcohol OR drugs

This is corrected in the revised manuscript.

P5, line 31: is divorce considered a social disadvantage?

Thank you for bringing this unfortunate wording to our attention. We have rephrased the sentence.

P5, line 36: Need to mention here the current legislation for DUI in Norway, e.g. the BAC limit and which other drugs (if any) are restricted

*We agree that the reader should be introduced to the current legislation for DUI in Norway earlier in manuscript and so we have moved the first paragraph under **Setting** in the Methods section to the Introduction.*

P5, line 39: How many people were tested in this study? Which year/s was it conducted?

We have elaborated on this in the Introduction.

P6, line 3: Are there countries where it is considered socially acceptable?

Thank you for this comment. We have rephrased this sentence to "It has been suggested that in countries with a strong law enforcement and strict punishments for DUI, people driving under the influence are a more selected group with higher incidence of substance use disorders (SUDs) along with higher levels of sensation-seeking and risk-taking behaviour."

P6, line 22: ...mortality is warranted for this group.' (rather than 'has been warranted')

Corrected accordingly

P6, line 35: This paragraph from 'Using data....observation period' should be moved to Methods.

We agree that parts of this paragraph belongs in the Methods section and have moved this to the first paragraph under the Measures heading.

P7, line 12: Does it contain data on persons imprisoned, or those that were released? Or both?

Thank you for notifying that clarification was needed. It contains both. We have added more information in the Methods section of the manuscript to clarify this.

P7, line 19: Is this a personal identification number, like a social security number? This may not be clear to foreign readers, need to explain how a PIN works in this context.

We have clarified that the PIN is a unique identification number given to persons who are born in Norway, have a valid residence permit for at least six months or have officially moved to Norway.

P7, line 19: Clarification is needed regarding the time of the offence/s relative to the death. Are these convictions at any time in the persons lifetime? Were cases only extracted when there was a match in the linkage between the deaths and the prison records?

Thank you for the notification. We have elaborated on this and hopefully made it clearer to the reader.

P7, line 24: spell out acronym for first time if haven't already

This is corrected in the revised manuscript.

P7, line 30: which drugs/drug classes are included? Just name a few to indicate the type of psychoactive substances- is cannabis included? Amphetamines?

We have added the requested information

P7, line 48: Some of this info on the Norway prison systems should be moved earlier to the introduction- it's very interesting and quite unique compared to other prison systems, and provides some useful background for the study.

We have moved the paragraphs under Setting to the be a part of the Introduction section as suggested.

P7, line 54: Is this different to nPRIS? Combine all data sources into this section.

Thank you for the suggestion. We have moved details about the data sources to the first paragraph in the Methods.

P8, line 16: Perhaps tabulate these data sources and specify exactly what data you retrieved from each

Thank you for the suggestion. We chose to keep it in the text because the manuscript already contains 4 tables.

P10, line 33: were they recidivist offenders, i.e. for the same crime? Or were the multi imprisonments for different crimes?

We have rephrased and added some information to clarify.

P11, line 48: suggest 'natural causes' instead of 'internal causes'

We have changed from internal and external causes of death to natural and unnatural.

P12, line17: than

This is corrected in the revised manuscript.

P13, line 58: Was there any association with recidivist offending or multiple convictions and mortality?

Thank you for this comment. Number of convictions have now been included in the regression and results have been discussed accordingly.

P14, line 3: remove apostrophe here- should read 'ratios'

This is corrected in the revised manuscript.

P14, line 19: Need to note significance level e.g. $p < 0.05$?

This is corrected in the revised manuscript.

P15, line 15: please define SUD

Substance use disorders (SUDs) is now defined where it is first mentioned in the Introduction section.

P15, line 20: convictions

This is corrected in the revised manuscript.

P15, line 32: Is this a lifetime risk with death occurring any time after a conviction?

The estimated risk of death is limited to the observation period. We have added this information in the manuscript.

P15, line 37: citation

This has been added

P15, line 51: suggest referring to median instead of means

Thank you for the suggestion. The data on the age of death were close to normally distributed and therefore we have chosen to use the mean.

P16, line 7: isn't this consistent with increasing natural disease with ageing?

Thank you for commenting on this. Yes, we expect that increased mortality from natural causes increase with age. However, in this population the risk of death from natural causes is high even at relatively young age and this is likely linked to a high prevalence of SUDs in this population. As we do

not have information on the prevalence of SUDs in our data we referred to the work of Nordentoft et al.

P16; line 14: Unclear of the relevance of this reference, suggest citing something without other comorbidities with a high mortality risk, like psychiatric disorders.

Please see the answer above. We have rephrased the paragraph to try to clarify.

P16; line 34:

Thank you for this comment. We have have elaborated on this in the 4th paragraph in the Discussion section

P16, line 44: suggests

This is corrected in the revised manuscript.

P17, line 46: with respect to

This is corrected in the revised manuscript.